# Phosphazene Functionalized Silsesquioxane-Based Porous Polymer as Thermally Stable and Reusable Catalyst for Bulk Ring-Opening Polymerization of ε-Caprolactone

**DOI:** 10.3390/polym15051291

**Published:** 2023-03-03

**Authors:** Yuliya A. Piskun, Evgenii A. Ksendzov, Anastasiya V. Resko, Mikhail A. Soldatov, Peter Timashev, Hongzhi Liu, Irina V. Vasilenko, Sergei V. Kostjuk

**Affiliations:** 1Research Institute for Physical Chemical Problems of the Belarusian State University, 14 Leningradskaya St., 220006 Minsk, Belarus; 2Department of Chemistry, Belarusian State University, 14 Leningradskaya St., 220050 Minsk, Belarus; 3Department of Science and Technology Projects, D. Mendeleev University of Chemical Technology of Russia, 9 Miusskaya Sq., 125047 Moscow, Russia; 4Institute for Regenerative Medicine, Sechenov University, 8-2 Trubetskaya St., 119991 Moscow, Russia; 5World-Class Research Center “Digital Biodesign and Personalized Healthcare”, Sechenov University, 8-2 Trubetskaya St., 119991 Moscow, Russia; 6Chemistry Department, Lomonosov Moscow State University, Leninskie Gory 1, 119991 Moscow, Russia; 7Key Laboratory of Special Functional Aggregated Materials, Shandong University, 27 Shanda Nanlu, Jinan 250100, China

**Keywords:** ring-opening polymerization, phosphazene base, ε-caprolactone, porous polymeric material, heterogeneous catalysis

## Abstract

The bulk ring-opening polymerization (ROP) of ε-caprolactone using phosphazene-containing porous polymeric material (HPCP) has been studied at high reaction temperatures (130–150 °C). HPCP in conjunction with benzyl alcohol as an initiator induced the living ROP of ε-caprolactone, affording polyesters with a controlled molecular weight up to 6000 g mol^−1^ and moderate polydispersity (Ð~1.5) under optimized conditions ([BnOH]/[CL] = 50; HPCP: 0.63 mM; 150 °C). Poly(ε-caprolactone)s with higher molecular weight (up to M_n_ = 14,000 g mol^−1^, Ð~1.9) were obtained at a lower temperature, at 130 °C. Due to its high thermal and chemical stability, HPCP can be reused for at least three consecutive cycles without a significant decrease in the catalyst efficiency. The tentative mechanism of the HPCP-catalyzed ROP of ε-caprolactone, the key stage of which consists of the activation of the initiator through the basic sites of the catalyst, was proposed.

## 1. Introduction

(Bio)degradable and biocompatible polyesters have been of significant interest during the last few decades as an alternative to traditional fossil-based plastics [1,2,3,4]. Although several microorganisms can directly degrade polyesters, their (bio)degradation mainly proceeds via hydrolytic scission of the polymer chain; the hydrolysis is also main pathway of degradation of polyesters in living bodies to non-toxic hydroxy acids [5,6,7,8]. Therefore, (bio)degradable and biocompatible polyesters have a variety of applications ranging between packaging and agricultural applications and biomedical and pharmaceutical applications [3,9,10,11]. Polyesters are usually synthesized via the ring-opening polymerization (ROP) of lactide, glycolide or lactones using alkoxides or complexes of different metals via the coordination-insertion mechanism [4,10,12,13]. Tin(II) 2-ethylhexanoate [12,14], which is currently used for the industrial production of polylactide and poly(ε-caprolactone), as well as aluminum triisopropoxide [13,14,15] and zinc lactate or stearate [16,17] were first reported as efficient catalysts for the synthesis of high molecular weight polyesters. However, the polymerizations with these catalysts are typically accompanied by undesirable side-reactions (inter- and intra-molecular transesterifications), leading to polymers with relatively high polydispersity. In this respect, the metal complexes with specially designed ligands afforded well-defined polyesters with tuned molecular weight, low polydispersity and high chain-end fidelity [4,12,13]; among these are the complexes of aluminum [12,18,19,20,21,22], titanium and zircoinum [23,24,25,26], rare earths metals [27,28], indium [29], yttrium [30], zinc [31] and many others [4,12,13,31]. Despite the fact that these catalytic complexes show high activity and stereoselectivity in the ROP of lactide and lactones and allow the preparation of different macromolecular architectures (block-, graft-, star-shaped copolymers), they are characterized by several disadvantages. First of all, these organometallic complexes are typically expensive due to the multistep synthetic procedures used for the preparation of the corresponding ligands. In addition, some metals can possess certain toxicity that requires the purification step for polyesters, particularly when used for medical purposes [4,12,13,27,31].

Organocatalysis is emerging as a powerful alternative to more traditional metal-based catalysts. Organic catalysts for the ROP of lactide and lactones are considered to be more environmentally friendly and less toxic, as well as being commercially available compared to most transition metal complexes. In addition, organic catalysts are well suited to a wide range of reaction conditions, solvents and monomers, and as a consequence of their acidic or basic nature, are usually very easy to remove from the resulting polymers by simple washing or trapping by the resin beads [32,33,34,35]. For the ROP of lactide, strong bases such as amidines and guanidines [36,37,38] were used in conjunction with the corresponding alcohols as initiators. The most frequently studied catalysts among them are 1,8-diazabiclo [5.4.0]undecene-7 (DBU), 1,5,7-triazabicyclo [4.4.0]-decene-5 (TBD) and 1,5,7-triazabicyclo [4.4.0]-7-methyldecene-5 (MTBD) [32,34,36,37,38]. In contrast, for the ROP of lactones, Bronsted acids such as trifluorormethanesulfonic acid, methanesulfonic acid or diphenylphospate are typically used, since strong bases mentioned above have shown quite low activity in the ROP of δ-valerolactone or ε-caprolactone [32,34,39,40].

In this respect, phosphazenes such as 2-tert-butylimino-2-diethylamino-1,3-dimethylperhydro-1,3,2-diazaphosphorine (BEMP) and N′-tert-butyl-N,N,N′,N′,N″,N″– hexamethylphosphorimide triamide (P1-t-Bu) represent an interesting class of organic catalysts capable of efficiently initiating the ROP polymerization of lactide and lactones [34,41,42,43,44,45]. Considering that most of the basic organic catalysts reported today suffer from low thermal stability and cannot be used under industrially relevant temperatures (~180 °C) [46,47], the high thermal stability of phosphazenes make them promising catalysts for the bulk ROP of lactide and lactones [43,48]. Another important feature of phosphazenes is the possibility to prepare a supported catalyst using polystyrene beads [49] or a porous polymeric aromatic framework [50]. These heterogeneous catalysts induced a controlled bulk ROP of δ-valerolactone or ε-caprolactone at elevated temperatures (100–110 °C), affording well-defined polyesters with M_n_ ≤ 10,000 g mol^−1^ and Ð ≤ 1.4 in 24–48 h. It was demonstrated that both of these catalysts can be separated from the reaction mixture and reused; they maintained the activity after at least four polymerization cycles [49,50].

Porous polymeric materials have been widely studied as efficient adsorbents, luminophores, sensors and heterogeneous catalysts for various reactions [51,52]. Hybrid porous polymers based on cage-like organosiloxanes represent an important subclass of porous materials, which can be used as reusable heterogeneous catalysts for different reactions [53,54]. Recently, we have synthesized several porous polymers based on cage-like silsesquioxane and phosphazene building blocks and have shown their efficiency not only as adsorbents, but also as a catalyst for the Knoevenagel reaction [55,56,57].

In this work, we report for the first time the bulk ROP of ε-caprolactone using a phosphazene-silsesquioxane-based porous polymer as a heterogeneous catalyst at 130–150 °C, taking advantage of its high thermal stability. Poly(ε-caprolactone)s with a controlled molecular weight up to M_n_ ≤ 14,000 g mol^−1^ and moderate polydispersity (Ð ≤ 1.7) were obtained under optimized conditions. We show here that the catalyst was quantitatively recovered from the reaction mixture and re-used for an additional three polymerization cycles (Figure 1).

## 2. Materials and Methods

### 2.1. Materials

All manipulations were carried out using the standard Schlenk techniques under an atmosphere of argon. ε-Caprolactone (CL, Acros, 99%, Saint Louis, MO, USA) and benzyl alcohol (BnOH, Sigma-Aldrich, 99.8%, Saint Louis, MO, USA) were dried over CaH_2_, distilled from CaH_2_ under reduced pressure and stored under argon. The catalyst (HPCP) was synthesized according to previously reported work [56], was washed with 0.1 M NaOH and distilled water and purified by dialysis. Finally, it was dried in vacuum at 50 °C overnight. Deuterated solvent CDCl_3_ (99.8%, for spectroscopy, Merck, Darmstadt, Germany), CH_2_Cl_2_ (Sigma-Aldrich, >99.5%, USA), tetrahydrofuran (LiChrosolv^®^, Merck, Darmstadt, Germany, >99.9%) were used as received. 

### 2.2. Polymerization Procedures

The ring-opening polymerization of ε-caprolactone in bulk was carried out as follows: ([monomer]/[BnOH] = 50, catalyst (HPCP) = 6.36 mM): 10 mL reactor (Schlenk tube) equipped with a magnetic stirrer bar was charged by 88.5 mg the catalyst (HPCP), immersed into an oil bath preheated to 40 °C for draying for 2 h in vacuum. After that, benzyl alcohol (0.09 mL, 176 mM) and ε-caprolactone (5 mL) were added to the reactor. Then, a Schlenk tube was immersed into an oil bath preheated to 130 °C for polymerization. For the kinetic experiments, 0.3 mL aliquots were withdrawn during the polymerization from the flask and subjected to ^1^H NMR spectroscopy to determine the monomer conversion and molecular weight of the produced polymers, respectively. The product was purified from traces of the catalyst through dissolving the polymer in CH_2_Cl_2_ and decanting its solution. 

### 2.3. Catalyst Recycling

After polymerization, the Schlenk tube was cooled to room temperature, filled with tetrahydrofuran; then, the catalyst was decanted and separated from the solution of poly(ε-caprolactone). The precipitate was washed several times by tetrahydrofuran. The resulting regenerated catalyst was dried in a vacuum oven for 12 h at 25 °C or 50 °C. Then, the catalyst was used for the ROP of ε-caprolactone according to the procedure described in Section 2.2.

### 2.4. Instrumentation

The number-average molecular weight and polydispersity of the polymers were determined by size exclusion chromatography (SEC). Measurements were performed using an Ultimate 3000 (Thermo Fisher Scientific Dionex, Sunnyvale, CA, USA) device with a PLgel MIXED-C column (7.5 mm × 300 mm, particle size 5 μm, Agilent) column and one precolumn (PLgel 5 μm Guard, Agilent Technologies, Santa Clara, CA, USA) thermostated at 30 °C. GPS traces were achieved using a refractometer or a UV-detector (λ = 255 nm). Tetrahydrofuran was used as a mobile phase with a flow rate of 1.0 mL min^−1^. The molecular weights and polydispersity were calculated based on the polystyrene standards (EasiCal, Agilent Technologies, Santa Clara, CA, USA) with M_w_/M_n_ ≤ 1.05 and using the Chromeleon 7.0 program (Thermo Fisher Scientific Dionex, Sunnyvale, CA, USA). The ^1^H NMR spectra were recorded in CDCl_3_ at 25 °C on a Bruker AC-500 spectrometer (Billerica, MA, USA) calibrated relative to the residual solvent resonance. The Fourier-transformed infrared (FT-IR) spectra were recorded using a Bruker TENSOR-27 infrared spectrophotometer (Billerica, MA, USA) from 4000 cm^−1^ to 400 cm^−1^ at a resolution. The solid-state ^13^ C CP/MAS NMR, ^29^Si MAS NMR and ^31^P MAS NMR spectra were recorded on a Bruker AVANCE-500 NMR (Billerica, MA, USA) spectrometer operating under a magnetic field strength of 9.4 T. The resonance frequencies at this field strength were 125, 99 and 202.6 MHz for ^13^ C NMR, ^29^Si NMR and ^31^P NMR, respectively. A chemagnetics 5 mm triple-resonance MAS probe was used to acquire ^13^C, ^29^Si and ^31^P NMR spectra. ^29^Si NMR spectra with high power proton decoupling were recorded with a π/2 pulse length of 5 μs, a recycle delay of 120 s, and a spinning rate of 5 kHz. Thermal gravimetric analysis (TGA) was performed on a Mettler-Toledo SDTA-854 (Greifensee, Switzerland) under a N_2_ atmosphere at a heating rate of 10 °C min^−1^ between room temperature and 800 °C. The elemental analysis of HPCP was conducted using an Elementar Vario EL III elemental analyzer (Germany) to give the following values: C: 38.94%; H: 3.981; N: 1.51%. From the elemental analysis data, the content of the potential basic sites in HPCP was calculated based on the content of nitrogen to be 1.07 mmol/g [49]. Then, assuming that only three nitrogen atoms from nine of the hexapyrroylcyclotriphosphazene unit in HPCP (see structure in Figure 1) are basic enough to induce the ROP; the content of the active sites was finally calculated to be 0.356 mmol/g.

## 3. Results and Discussions

### 3.1. HPCP-Catalyzed ROP of ε-Caprolactone

The phosphazene-silsesquioxane-based porous polymer (HPCP, see structure on Figure 1) was prepared through the cross-linking reaction between octavinylsilsesquioxane and hexapyrroylcyclotriphosphazene via the Friedel-Crafts reaction, according to the procedure reported previously [55,57]. The FTIR spectrum of HPCP exhibits broad peaks at 1300–1000 cm^−1^, corresponding to the Si-O-Si and P-N bonds (Figure A1). In the ^13^C NMR spectrum of the HPCP, the broad peaks at 34 ppm are attributed to carbon atoms of single C-C bonds (Figure A2), formed from the vinyl group of octavinylsilsesquioxane in the course of the Friedel-Crafts reaction. The signals in the range of 110–140 ppm are attributed to the carbon atoms of the pyrrolyl rings and unreacted vinyl groups. In the ^29^Si NMR spectrum, there are two peaks at 66 and 79 ppm, which are attributed to the T_3_ units (T_n_: -C-Si(OSi)_n_(OH)_3-n_) and unreacted vinyl groups (CH_2_ = CH-Si≡), respectively (Figure A3). No signals of the T_1_, T_2_ and Q_n_ units are observed, suggesting that no cages collapsed during the reaction. The ^31^P NMR spectrum exhibits only one signal at 4 ppm (Figure A4), which shows that no side reaction in the phosphazene rings, such as ring-opening or resubstitution, took place. Taking into account the high thermal stability (Figure A5) and micro- and meso-porous structure of HPCP [55,57], it can be considered to be a promising heterogeneous catalyst for bulk ROP.

In the first series of experiments, we investigated the effect of the temperature on the bulk ROP of ε-caprolactone catalyzed by HPCP in the presence of benzyl alcohol as an initiator (Figure 1, Table 1). 

The polymerization is rather slow at 100 °C, even at a relatively high content of HPCP, affording the polyesters with a lower than theoretical molecular weight (run 1, Table 1). The increase in the reaction temperature to 130 °C in conjunction with the decrease in the catalyst concentration allowed the preparation of the poly(ε-caprolactone) with a M_n_ close to theoretical one and a relatively high degree of number-average functionality on the benzyl head group (F_n_) (see runs 1–3 in Table 1). The closeness of the values of M_n_ determined by the SEC and NMR indicates that, on one hand, the formation of macrocycles is not significant under the investigated conditions ([CL]/[BnOH] = 50; HPCP: 6.36 mM, 130 °C, run 3 in Table 1). On the other hand, F_n_ is systematically lower than 100%, indicating the simultaneous initiation of polymerization by protic impurities such as H_2_O (vide infra). The last assumption is confirmed by the polymerization experiment performed without an addition of benzyl alcohol, which leads to quantitative polymerization in 48 h (although at ten times higher catalyst content), with the generation of poly(ε-caprolactone) with M_n_ = 14,200 g mol^−1^ and moderate polydispersity (Ð = 1.90) (run 6, Table 1). It should be noted that the decrease in the benzyl alcohol concentration from [CL]/[BnOH] = 50 to [CL]/[BnOH] = 500 results in the significant retardation of the polymerization of ε-caprolactone performed at 130 °C, affording polymers with a slightly higher molecular weight and lower polydispersity (runs 3 and 4 in Table 1). However, the synthesized polymers displayed quite low functionality, while the experimental values of M_n_ are much lower that theoretical ones, indicating that at a low initiator concentration, the initiation by protic impurities becomes predominant. Aiming at increasing the reaction rate, the polymerization was conducted at a higher temperature (150 °C) to afford the poly(ε-caprolactone) in quantitative conversion in 72 h. In addition, the synthesized polymers are characterized by higher F_n_ than those prepared at 130 °C, pointing out that the initiation from benzyl alcohol is more efficient at a higher temperature (runs 4, 5 in Table 1). Therefore, the bulk ROP of ε-caprolactone was further studied at 150 °C.

Indeed, the increase in the temperature results in much faster polymerization, even in the absence of BnOH, although at the expense of decreasing the molecular weight (compare run 6 in Table 1 and run 7 in Table 2). The last observation could be explained by the more efficient initiation by the protic impurities at a higher temperature that is, to some extent, in contradiction with the previously observed, higher F_n_ for the poly(ε-caprolactone) prepared at 150 °C compared to that synthesized at 130 °C (runs 4, 5 in Table 1). This contradiction may be explained by the higher concentration of catalyst used in the experiments without the addition of BnOH (run 6 in Table 1 and run 7 in Table 2) compared to the experiments with an initiator (runs 4, 5 in Table 1). Therefore, the catalyst by itself can contain the protic impurities, which are difficult to remove due to the relatively high basicity and porous structure of the HPCP [55,57]. In this work, we washed the catalyst with water (see Section 2.1 for details) in order to remove the acidic impurities that arose in the course of the catalyst preparation [55,57]. In fact, the use of unpurified HPCP induced a faster ROP of ε-caprolactone, but afforded the polyesters with higher polydispersity (runs 8, 9 in Table 2).

Evidently, the decrease in the catalyst concentration leads to poly(ε-caprolactone) with lower polydispersity and higher functionality at both of the temperatures studied (see runs 2, 3 in Table 1 and runs 8, 11 in Table 2). At a low catalyst concertation, the use of an initiator is required due to the extremely slow polymerization under BnOH-free conditions (run 10, Table 2). In addition, the polyesters synthesized without the addition of an initiator are typically characterized by high polydispersity, displaying non-symmetrical and often bimodal SEC curves (curve 1 in Figure 1). The molecular weight distribution becomes narrow at a higher temperature (150 °C) compared to 130 °C due to the formation of shorter chains, but the shoulder in the high molecular weight region is still visible (curve 2 in Figure 1). A similar bimodality is often observed for poly(ε-caprolactone) with low functionality (curve 3 in Figure 1), which were obtained at relatively high [CL]/[BnOH] ratios, due to the simultaneous initiation by benzyl alcohol and H_2_O. The poly(ε-caprolactone)s with lower polydispersity and higher functionality were obtained at the lowest HPCP concentration and a low [CL]/[BnOH] ratio (run 12 in Table 2 and curve 4 in Figure 1).

### 3.2. Controlled ROP of ε-Caprolactone

In order to confirm the controlled nature of the ROP of ε-caprolactone catalyzed by HPCP, the kinetics for polymerization with and without an initiator (BnOH) were briefly investigated. The first-order plots are linear and passed through zero for all of the polymerization experiments (Figure 2a). The rate of the HPCP-catalyzed polymerization with BnOH was higher than that without an addition of an initiator (k_p,app_ = 9.9 × 10^−2^ h^−1^ and 6.5 × 10^−2^ h^−1^, respectively), indicating that an initiation by alcohol is more efficient than by protic impurities. The number-average molecular weight increased with the increasing monomer conversions for both [CL]/[BnOH] ratios studied here, pointing out that, on one hand, the HPCP-catalyzed ROP of ε-caprolactone proceeds in a controlled fashion (Figure 2b). On the other hand, the good correlation between the experimental values of the M_n_ and theoretical line is observed only for the polymerization performed at a low [CL]/[BnOH] ratio, while the significant deviation from the theoretical M_n_ was observed for the polyesters prepared at [CL]/[BnOH] = 500 (M_n,theor_ = 57,000 g mol^−1^ for complete monomer conversion). More surprisingly, the experimental values of M_n_ are at the same theoretical line for the poly(ε-caprolactone) obtained without benzyl alcohol as the polymers obtained at [CL]/[BnOH] = 50 and 500, respectively (Figure 2b). Considering that a higher molecular weight (M_n_ = 14,200 g mol^−1^) can be obtained only under alcohol-free conditions at a lower temperature (130 °C), the above-mentioned observation could be explained by the significant number of protic impurities trapped by the basic active species of catalyst. The initiation by protic impurities becomes significant at a high reaction temperature, which results in the reduction of the molecular weight. This explanation is confirmed by the relatively low functionality of the polyesters prepared at a high [CL]/[BnOH] ratio and a high temperature (see Table 2). This indicates that most of the chains are generated due to the invitation by adventitious H_2_O, but not from benzyl alcohol. To validate this concept further, the chain-end structure of poly(ε-caprolactone), prepared with and without benzyl alcohol, was analyzed by ^1^H NMR spectroscopy. 

### 3.3. Polymer Characterization

The typical ^1^H NMR spectrum of crude poly(ε-caprolactone) prepared with a BnOH/HPCP initiating system based is shown in Figure 3a. The ^1^H NMR spectrum of the poly(ε-caprolactone) purified via re-precipitation in methanol is presented in Figure A6. Among the large absorptions of the main-chain methylene protons at 1.38 (c), 1.77 (b + d), 2.30 (a) and 4.06 (e) ppm, the less intensive resonance at 3.64 (f) ppm attributed to the hydroxylmethylene end group is also detected. In addition, the signals of the methylene and aromatic protons of the benzyl head group appear at 5.11 (g) ppm and 7.35 (h + k), respectively. The peaks marked with dashes (a’–e’) belong to the monomer protons of the unreacted ε-caprolactone (Figure 3). Based on the ^1^H NMR spectrum of the unpurified poly(ε-caprolactone), the conversion was calculated according to the following equation: Conv. = I(e)/(I(e) + I(e’)) × 100%. The ^1^H NMR spectrum of the poly(ε-caprolactone) prepared with a BnOH/HPCP initiating system was also used for the calculation of the number-average molecular weight (M_n_ = I(e)/I(f) × 114 + 108, where 114 is the molar mass of ε-caprolactone and benzyl alcohol, respectively) and the number-average functionality of the benzyl group (F_n_ = I(g)/I(f) × 100) (see Figure 3a for details). 

The ^1^H NMR spectrum of the poly(ε-caprolactone) synthesized via the HPCP-catalyzed ROP of ε-caprolactone without the addition of an initiator is presented in Figure 3b. Among the signals of the main-chain protons, the signal of the hydroxylmethylene end group is detected at 3.64 ppm (f). The initiation of the ROP of ε-caprolactone would theoretically lead to the generation of the carboxylic acid head group. However, this group is typically not visible in ^1^H NMR spectroscopy [58]. Therefore, considering this fact and taking into account the only slightly higher values of M_n_ (NMR) compared to M_n_ (SEC) for the polymers obtained without the addition of an initiator, we can conclude that initiation by protic impurities is indeed the main pathway for the initiation of the HPCP-catalyzed ROP of ε-caprolactone without the addition of an initiator.

### 3.4. Catalyst Recycling

Aiming at exploiting the advantage of a heterogeneous catalysis, the possibility of the separation and reuse of the HPCP in the ROP of ε-caprolactone was briefly estimated. As it was mentioned in the Experimental section (Section 2.3), the catalyst could be easily separated from the reaction mixture by simple decantation and washing by tetrahydrofuran, and then reused for the ROP of ε-caprolactone after drying in vacuum for 12 h at 25 °C. It should be noted that no significant loss of the catalyst was observed in the course of catalyst separation. As shown in Table 3, the HPCP has been reused for at least three consecutive cycles without a significant decrease in the efficiency. Some loss of catalyst activity during its reuse may be explained by the increase in the fraction of protic impurities coordinated to the basic active species of HPCP. This assumption is consistent with the decrease in the molecular weight and functionality with the increasing number of cycles of catalyst reuse (Table 3). This point was confirmed by the increase in the temperature of the catalyst drying, from 25 to 50 °C, after its recovery from the reaction mixture. At these conditions, the conversion, molecular weight and functionality almost do not change with the increasing number of cycles of catalyst reuse (Table 3). Thus, these preliminary results demonstrated that the phosphazene-silsesquioxane-based porous polymer (HPCP) is indeed reusable and a promising catalyst for the green heterogeneous ROP of ε-caprolactone. 

## 4. Discussion

Based on the obtained results as well as taking into account the previous work on the use of HPCP as a reusable catalyst of the Knoevenagel reaction [57], we proposed the following mechanism of the HPCP-catalyzed ROP of ε-caprolactone (Figure 2). At the initiating step of the polymerization, the initiator (benzyl alcohol, adventitious H_2_O or oligo(ε-caprolactone)) is activated through the coordination with the basic active species of HPCP. The monomer insertion occurs through the nucleophilic attack of the activated initiator onto the carbonyl oxygen of ε-caprolactone, followed by the formation of an adduct of the initiator with the monomer and the regeneration of the active center. The chain growth proceeds via the multiple insertion of the monomer into the CH_2_–OH bond of the activated hydroxyl-terminated oligo(ε-caprolactone) generated at the earlier steps of the ROP of ε-caprolactone (Figure 2). 

As HPCP contains traces of protic impurities, the simultaneous initiation of the ROP of ε-caprolactone by H_2_O and benzyl alcohol occurred. The efficiency of this undesirable initiation increased with the increase in the temperature of polymerization or the decrease in the concentration of benzyl alcohol. This, in turn, led to a decrease in the number-average molecular weight of the synthesized polyesters. The highest initiation efficiency by benzyl alcohol and the lowest polydispersity was achieved at the lowest catalyst concentration (see Table 2), confirming the presence of protic impurities in the structure of the catalyst.

In summary, the phosphazene-silsesquioxane-based porous polymer (HPCP) is a promising heterogeneous catalyst for the bulk ROP of cyclic esters due to its high thermal and chemical stability, which allow both polymerization at high temperatures, meeting the requirements of the industry, and the reuse of the catalyst without any significant loss in its activity. The microporous structure of the catalyst allows, to some extent, the suppression of the undesirable transesterification reactions, resulting in poly(ε-caprolactone) with moderate polydispersity (Ð = 1.5–1.7), even at such a high reaction temperature as 150 °C.

## 5. Conclusions

In conclusion, the new promising heterogeneous catalyst for the ROP of cyclic esters based on the phosphazene-silsesquioxane-based porous polymer was tested in this study. This catalyst, due to its high thermal stability, induced the bulk ROP of ε-caprolactone at a high reaction temperature (130 °C–150 °C) and can be easily separated from the reaction mixture and reused for at least three consecutive cycles without any significant loss of activity. The bulk ROP of ε-caprolactone with a BnOH/HPCP initiating system proceeds in a controlled fashion, affording polyesters with a M_n_ up to 6000 g mol^−1^ and moderate polydispersity (Ð~1.5) under optimized conditions ([BnOH]/[CL] = 50; HPCP: 0.63 mM; 150 °C). Poly(ε-caprolactone)s with a higher molecular weight (up to M_n_ = 14,000 g mol^−1^) were obtained at a lower temperature (130 °C) without the addition of an initiator due to the low efficiency of the initiation by protic impurities under such conditions. Thus, these preliminary results demonstrated that the phosphazene-silsesquioxane-based porous polymer (HPCP) is indeed reusable and is a promising catalyst for the green heterogeneous ROP of ε-caprolactone. Aiming at broadening the scope of the usage of this catalyst, the ROP of other lactones, including δ-valerolactrone, is under the investigation in our lab. In addition, the further design of the structure of the catalyst is also required in order to increase its activity.

## Data Availability

Not applicable.

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
