# Peer review of "Phosphazene Functionalized Silsesquioxane-Based Porous Polymer as Thermally Stable and Reusable Catalyst for Bulk Ring-Opening Polymerization of ε-Caprolactone"

_polymers, 2023, doi:10.3390/polym15051291_

Round 1

Reviewer 1 Report

Attached

Reviewer 2 Report

This manuscript by Piskun and coworkers reports bulk ring-opening polymerization (ROP) of ε-caprolactone using a heterogeneous organocatalyst phosphazenecontaining porous polymeric material (HPCP). Due to its high thermal and chemical stability,  HPCP can be reused for at least three consecutive cycles at high temperature (150 °C) without significant decrease of catalyst efficiency, although the Mns of resultant polymers become significantly lower. This manuscript need to be deeply improved before publication for the following reasons:

(1) There is significant flaw in this manuscript. The authors do not under the living polymerization, which should produce polymers with predicable Mns and narrow distribution (usually < 1.20).

(2) In some cases, the experimental Mns by GPC and NMR are far lower than theoretical values, which is barely explained by the presence of trace of protic impurity. Therefore, MALDI-TOF analysis for resultant polymers is recommended in order to analyze the end groups.

(3) In Figure 1a, some plots are based on two or three points, which is insufficient and meaningless.

(4) Figure 2 is for the crude mixture of polymerization reaction rather than for PCL synthesized.

(5) Although HPCP can be reused without significant decrease of activity, the Mns of resultant polymers become significantly lower. The authors need to make the comment on this issue.

Reviewer 3 Report

The authors reported a recyclable metal-free porous polymer-based catalyst for ring-opening polymerization of ε-caprolactone. The catalytic efficiency and the recycling of the catalyst were promising. There are a few specific comments as follows.

1)      Scheme 1, shouldn’t the catalyst be on top of the reaction arrow and the initiator on the left side as a reactant?

2)      The appendix figures can be combined into one figure and inserted into the main text.

3)      Page 6, line 1, “The molecular weight distribution becomes narrow at higher temperature”, what is this compared to?

4)      Figure 3 seems redundant since the monomer conversions have been reported in Table 3. Essentially, this figure does not provide any other information.

5)      Perhaps it is worth mentioning that porous polymer-based recyclable catalysts are very attractive, but purifying the catalyst or evaluating the impurities' type and quantity seems more challenging. Thus, it might be a challenge for industry usage to get consistent and reproducible results. In addition, it seems difficult to get high molecular weight polymers. 

Round 2

Reviewer 2 Report

Thanks the authors for their response, and this reviewer believes this manuscript is now ready for publication.